# The invasive pathogen *Yersinia pestis* disrupts host blood vasculature to spread and provoke hemorrhages

Guillain Mikaty[1¤a]*, Héloïse Coullon[1¤b], Laurence Fiette[2¤c], Javier Pizarro-Cerdá[1], Elisabeth Carniel[1]

**1** Institut Pasteur, Yersinia Research Unit, Paris, France, **2** Institut Pasteur, Unité d'histopathologie humaine et modèles animaux, Paris, France

¤a Current address: Institut Pasteur, Environment and Infectious Risks research and expertise Unit, Laboratory for urgent response to biological threats (ERI-CIBU), Paris, France
¤b Current address: Washington University School of Medicine, Division of Infectious Diseases, Dept. of Medicine; St. Louis, Missouri, United States of America
¤c Current address: IMMR, Paris, France
* guillain.mikaty@pasteur.fr

**Data Availability Statement:** All relevant data are within the manuscript and its Supporting Information files.

## Abstract

*Yersinia pestis* is a powerful pathogen with a rare invasive capacity. After a flea bite, the plague bacillus can reach the bloodstream in a matter of days giving way to invade the whole organism reaching all organs and provoking disseminated hemorrhages. However, the mechanisms used by this bacterium to cross and disrupt the endothelial vascular barrier remain poorly understood. In this study, an innovative model of *in vivo* infection was used to focus on the interaction between *Y. pestis* and its host vascular system. In the draining lymph nodes and in secondary organs, bacteria provoked the porosity and disruption of blood vessels. An *in vitro* model of endothelial barrier showed a role in this phenotype for the pYV/pCD1 plasmid that carries a Type Three Secretion System. This work supports that the pYV/pCD1 plasmid is responsible for the powerful tissue invasiveness capacity of the plague bacillus and the hemorrhagic features of plague.

## Author summary

The plague bacillus, *Yersinia pestis*, is a powerful pathogen with a rare invasive capacity and is among the few bacteria capable to provoke disseminated hemorrhages. However, the mechanisms used by this bacterium to cross and disrupt the endothelial vascular barrier remain poorly understood. Recent technical progress in microscopy, associated with the use of original fluorescent mutant in mice, allowed us to develop an innovative model of infection *in vivo*. This model permitted to look directly into the interaction between *Y. pestis* and its host vascular system, in 3D reconstructed tissues without physical alteration. We were able to observe the degradation of blood vessels in the draining lymph nodes and to visualize the spreading of the bacteria into secondary organs directly through the vascular barrier. Classical *in vitro* experiments validated the *in vivo* observation and

**Funding:** GM received grants from the CEA (NRBC project #17.1). https://www.cea.fr/Pages/domaines-recherche/defense-securite/recherches-CEA-programme-NRBC-E.aspx The funders had no role in study design, data collection and analysis, decision to publish, or preparation of the manuscript.

**Competing interests:** The authors have declared that no competing interests exist.

demonstrated the role of some of the bacterial components in this phenotype. This work shows an unprecedented visualization of the pathogenesis of *Y. pestis* and decipher part of the powerful invasiveness capacity of the plague bacillus and the hemorrhagic features of plague.

## Introduction

Plague is an infection caused by the invasive Gram-negative bacillus *Yersinia pestis*. It is among the most dramatic bacterial diseases in human history [1]. In spite of its disappearance in most developed countries, plague still represents a significant public health problem in many regions of the world [2]. Madagascar is currently the most active plague focus worldwide, with hundreds of human cases every year and regular epidemics [3]. Since the 1990's, the disease has also re-emerged in countries where it was thought to be extinct [4–6]. Furthermore, the discovery of natural isolates of *Y. pestis* carrying antibiotic-resistance plasmids [7,8] associated with the possibility of using *Y. pestis* as a bioweapon in the international context of terrorism is also of global concern.

Bubonic plague, the most common clinical presentation in humans and the usual form of plague in the rodent reservoir, occurs after an infectious fleabite. Studies in animal models have shown that *Y. pestis* may remain for various periods of time at the site of inoculation in the dermis and multiply locally. Afterwards the bacteria are drained through the lymphatic flux to the proximal lymph node where they form the pathognomonic bubo, and then to the secondary ipsilateral lymph node [9–11]. Inside the lymph nodes, *Y. pestis* enters through the subcapsular sinus, where the bacteria replicate and spread within the sinus. Bacteria spread into the cortex where they multiply, provoking the recruitment of numerous polymorphonuclear leukocytes. If the innate immune system of the lymph node is strong enough to contain the bacteria, the bubo suppurates and the patient recovers. However, in most instances the bacteria overpower the innate immune system of the lymph node and spread systemically [12,13]. The time elapsed between the colonization of the lymph node and the fatal outcome is very short (2.2 days on average) in the mouse experimental model [10]. Several hypotheses have been proposed for the mechanism used by *Y. pestis* to enter the bloodstream: (i) an invasive process by active degradation of the blood vessels of the node [11], (ii) a release of the bacteria carried by the lymphatic vessels into the blood through the subclavian vein, or (iii) a carriage of *Y. pestis* inside leukocytes from the lymph to the blood [14]. Once in the blood, bacteria are filtered by secondary lymphoid organs (spleen and liver), until the filtering capacity of these organs is overwhelmed, allowing the bacteria to spread and cause a severe and terminal septicemia, sometimes associated with internal and external bleedings. Massive and diffuse hemorrhages in all tissues including the lymph nodes, have been a striking feature of postmortem pathological examination of human plague victims [13,15]. Comparable hemorrhages were also reported in wild animals that succumbed to plague [16,17]. This feature of plague pathogenesis is experimentally reproducible in various animal models [11,12,18,19]. A classically accepted view is that a Disseminated Intravascular Coagulopathy (DIC) occurs during septicemic stages and provokes these hemorrhages [18,20,21]. However, DIC usually occurs late in the pathogenesis of Gram-negative bacteria whereas hemorrhages are observable as early as day 3 post-infection with *Y. pestis*. Thus suggesting that *Y. pestis* has the ability to alter blood vessel integrity independently of coagulation defect.

*Y. pestis* express various virulence factors that play important roles during the invasion and colonization of its hosts. The high-pathogenicity island, or HPI, carries essential virulence

genes involved in iron acquisition and is encoded on the chromosome [22]. Three plasmids carry specific virulence factors [23–24]: i) The pPla plasmid (also known as pPCP1) carries notably the plasminogen activator Pla, an adhesin with enzymatic activities capable of converting plasminogen to plasmin, thus degrading extracellular matrix and fibrin clots *in vivo*; ii) the pMT plasmid (also known as pFra) of *Y. pestis* carries the *caf 1* gene that encodes for the F1 pseudocapsule, involved in the resistance to phagocytosis [25]; iii) the pYV/pCD1 plasmid carries genes necessary to synthetize a Type Three Secretion System (TTSS) that allows the injection of effectors (Yops) inside the host cell cytosol [26]. These three plasmids are associated with *Y. pestis* virulence. The pYV/pCD1 plasmid is shared by and essential for the virulence of the three pathogenic *Yersinia* species, *Y. pestis*, Y. *pseudotuberculosis* and *Y. enterocolitica* [27]. They might be involved in the powerful invasiveness of *Y. pestis*. Although the capacity of *Y. pestis* to alter the vascular barriers may be important during the pathological process, the interactions of the pathogen with endothelial cells remain mainly unexplored. The aim of this study was, using *in vitro* and *in vivo* models of infection, to get insight into the processes that allow *Y. pestis* to cross the blood vessel barrier and ultimately cause hemorrhages. The results show that *Y. pestis* can degrade blood vasculature within the draining lymph node *in vivo*, certainly causing the internal hemorrhages frequently observable. Once in the bloodstream, bacteria can also spread systematically and cross the vascular barrier from the lumen to the organs, eventually degrading the tissues in the secondary organs. The *in vitro* model of infection of cellular vascular barrier highlights the central role of the pYV/pCD1 plasmid in this phenomenon.

## Materials and methods

### Ethics statement

All animals were housed in a level 3 animal facility accredited by the French Ministry of Agriculture (accreditation B 75 15–01), and were infected in compliance with French and European regulations (EC Directive 86/609, French Law 2001–486), following the approved protocol CETEA 2014-0025/MESR 008223 by the internal Institut Pasteur ethic board. For infection of Flk-1$^{GFP/+}$ mice, death of animals was never intended as outcome of the experiments and no mice died before planned sacrifice. For the LD50 measurement of the *Y. pestis* CO92 pFU96+ strain, death of animal was the planned outcome. Humane endpoint were defined and animals were monitored twice a day; animals appearing evidently moribund (apathetic, shivering, cold, etc.) were immediately sacrificed. EC and GM were trained by the Institut Pasteur internal training for handling and care of animals and for animal experimentation.

### Bacteria and culture

The *Y. pestis* strains used were CO92 wild type [28] and its pYV/pCD1-cured and Δ*caf* derivatives [29]; 6/69 wild type and its pPla-cured derivatives [30,31]. Plasmid pFU96 [32] that confers red-fluorescence was kindly provided by P. Dersch (department of Molecular Infection Biology, Helmholtz Centre for Infection Research, Braunschweig, Germany) and was introduced into *Y. pestis* CO92 by electroporation. For animal experiments, bacteria were cultured at 28˚C for 36h on Luria Bertani agar plates supplemented with 0.002% (w/v) hemin (LBH). For cell infection, bacteria were grown on LBH plates at 28˚C for 24h, and at 37˚C for another 12h. All experiments with live *Y. pestis* were performed in a biosafety level 3 laboratory.

### Animal experiments

Flk-1$^{GFP/+}$ mice [33] exhibiting a bright GFP signal in all endothelial cells due to the insertion of the gene encoding green fluorescent protein (GFP) into the VEGF receptor-2 gene locus,

were kindly provided by Alexander Medvinsky (Institute for Stem Cell Research, University of Edinburgh, UK). For infection, 100 μl of $5x10^3$ cfu/ml bacterial suspensions were injected subcutaneously into the right lateral ventral region. The LD50 of the *Y. pestis* CO92 pFU96+ strain was measured using six-week-old OF1 female mice (Charles River Laboratory) according to the method of Reed and Muench [34].

## Cell culture & infection

Human Dermal Microvascular Endothelial Cells (Promocell) were grown in Endothelial Cell Medium MV2 with supplements (Promocell) at 37˚C in 5% CO2. Five days prior to infection, cells were plated on glass coverslips in 24-well plates (for immunofluorescence experiments) or on Transwells (0.4 or 3 μm filters) coated with Type-I Collagen (R&D System) at a density of $0.25x10^5$ cells/well. Continuous cell monolayers ($\approx 5x10^5$ cells/well) were infected at a Multiplicity of Infection (MOI) of 100 for 30 min to allow bacterial adhesion. After washing with PBS and addition of fresh medium to the monolayer, the incubation was continued for at least 2h. The MOI and the timing were chosen following preliminary experiments in which a serial increasing MOI was used to infect HDMEC cells (1; 10; 50; 100; 1000) for various times (30', 60', 90, 120', 240'). In all conditions, the outcome was the same, only the percentage of rounding was varying depending on the MOI and time. Infection too long or with a MOI too strong resulted in detachment and/or death of the cells. MOI 100 for 150 minutes was found to be optimal for the observation of cell rounding with limited cell death/detachment, and was subsequently selected for further experiments. In a first set of permeability assays, 1 mg/ml of 4 kDa FITC-Dextran (Sigma aldrich) was added to the upper chamber of 0.4 μm Transwells covered with HDMEC that were either untreated (control) or infected for 2.5h. Every 20 min, 100 μl samples from the lower chamber were taken to measure optical density at 485 nm and 530 nm for FITC excitation and emission respectively. Fluorescence value was converted into protein mass using the fluorescence measured for a standard curve of 4 kDa FITC-Dextran. In another set of permeability assays to measure the capacity of *Y. pestis* to translocate through the cell monolayer, 3 μm Transwells were used. The bacteria in the upper chamber were not removed after 30 min, and 100 μl samples were taken from the lower chamber after 3.5h and 4.5h to enumerate cfu.

Bacterial filtrates were obtained from filtration (through at 0.22 μm filter) of suspensions of $5x10^7$ bacteria cultured in cell culture medium at 28˚C or 37˚C for 3 hours. Bacterial sonicates were obtained from sonication of $10^9$ bacteria cultured on Agar plate at 28˚C or 37˚C and resuspended in cold saline added with protease inhibitors (Roche, Complete) and PMSF (Sigma). Sonicates were filtrated through 0.22 μm filters. Cells ($2x10^5$ HDMEC) were incubated for 5 hours at 37˚C with the equivalent of $5x10^7$ bacterial filtrate or sonicate.

## Confocal microscopy, histology and immunofluorescence microscopy

Right inguinal lymph nodes were recovered as draining lymph node and left inguinal lymph nodes as secondary infected organs. Lymph nodes and spleen were fixed in 4% neutral buffered paraformaldehyde (PFA) for 48h, sliced with a 200 μm-thick setting using a vibratome (Leica VT 1200S), and mounted with Vectashield mounting medium (Vector Laboratories). Slides were observed and imaged with a spinning-disc Cellvoyager CV1000 confocal system (Yokagawa) at the Imagopole of the Center for Innovation & Technological Research from the Institut Pasteur. For histopathological examination, non-infected inguinal lymph nodes were taken and fixed in 4% neutral buffered formalin for 48h, and embedded in paraffin. Four μm tissue sections were stained with hematoxylin-eosin. Histology slides were examined through an Eclipse 5Oi Nikon microscope equipped with a DSRi1 camera (Nikon).

*In vitro*-infected cells were fixed in 4% PFA at specified times and exposed to the following primary antibodies for 1h: mouse anti-F1 monoclonal antibody (1:500; B18-1, [35]); rabbit *anti-Yersinia* polyclonal antiserum (1:1000; [12]), mouse anti-human VE-Cadherin monoclonal antibody (1:100; BV9, Abcam). Afterward, they were washed 3 times with PBS and incubated with the following secondary antibodies for 1h: Alexa Fluor 488 goat anti-mouse (1:500; Invitrogen) or anti-rabbit (1:1000; Invitrogen) IgG; rhodamin-phalloidin (1:200; Invitrogen), and DAPI (1:10.000, Interchim). After three washings with PBS, stained cells were mounted in moviol and observed under a confocal microscope LSM700 upright (Zeiss) coupled to a color camera. The average number of round cells per field was determined by counting the number of cells in at least three different fields per condition for each experiment. Cell fields were chosen randomly using the DAPI channel, then the field was switched to green and the presence of holes around the cell was enumerated. All the *in vitro* experiments were repeated independently at least three times. Images were processed and analyzed using Photoshop (Adobe) for immunofluorescence experiments and FIJI (ImageJ) for confocal experiments. No non-linear alterations were brought to the images.

## Results

### *Y. pestis*–infected lymph node display degraded blood vessels

Hemorrhages are a classical feature of plague infection frequently observed in buboes or organs [11,12,18,19]. However, the actual mechanism involved in the disruption of blood vessels was never properly determined using classical histopathology examination of stained tissue sections. Indeed, the interactions between bacteria and the damaged blood vessels are too localized to be observed by usual histopathological sections of 5–8 μm thickness. To address this question, an *in vivo* model dedicated to this interaction was developed. In this model, fluorescent bacteria were used to infect mice expressing GFP-tagged vasculature. This model allowed the qualitative observation of the direct interaction between bacteria and blood vessels with minimal treatments allowing tissue quality to be preserved.

First, the plasmid pFU96 encoding the Red Fluorescent Protein (RFP) was introduced into the fully virulent *Y. pestis* strain CO92 [32]. The strain was monitored for the expression of the virulence factors: pYV/pCD1, pPla, pFra and the HPI locus as described in [27]. The LD50, determined according to the method of Reed and Muench [34], was 16 bacteria, showing no significant difference with the parental strain CO92.

Second, this RFP-CO92 strain was used to infect Flk1-GFP transgenic mice that express a fluorescent blood vascular system due to their Green Fluorescent Protein (GFP)-tagged VEGF receptor [33]. Fig 1A presents the green vasculature of the inguinal lymph node of a non-infected Flk1-GFP mouse (right panel). The comparison to a classically hematoxylin-eosin stained histological section of inguinal node (center panel) shows the highly vascularized medulla and the cortex of the node. In the cortex, some spots present a less dense presence of blood vessels and correspond to follicles. The capsule is clearly visible around the node.

The Flk1-GFP mice were infected subcutaneously with 500 cfu of *Y. pestis* RFP-CO92. RFP-CO92 infection of the Flk1-GFP mice caused bacterial loads in their spleen similar to those observed in the usual OF1 mouse model infected with the parental CO92 strain (an average of $3.10^4$cfu/organ at 48hrs and $6.10^4$cfu/organ at 72hrs).

The inguinal draining lymph node of the Flk1-GFP mice infected was sampled on day 3 (D3). At this stage, classic histopathological observations allow to observe important masses of bacteria in the subcapsular sinus of the draining lymph node that spread through the cortex (S1 Fig). Bacteria are surrounded by infiltrated Polynuclear neutrophils. Different studies also described clear signs of hemorrhages observable in the lymph nodes of most infected mice

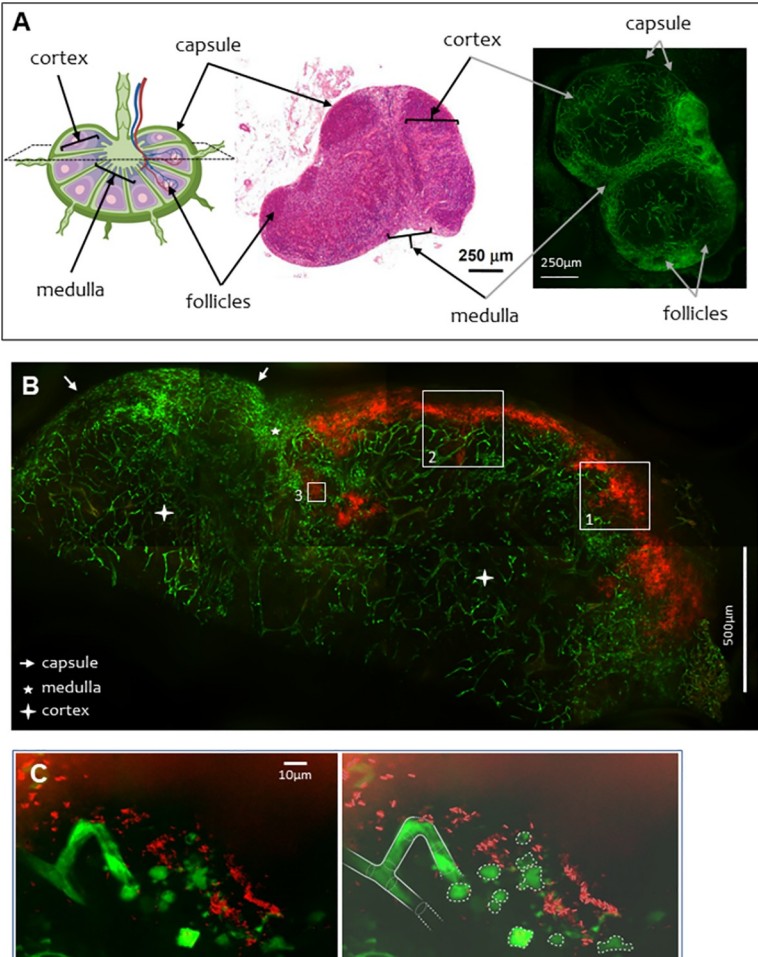

**Fig 1. Confocal microscopy, observation of the blood vessels in the draining lymph-node of non-infected and *Y. pestis*-infected Flk1-GFP mice. A. Left,** schematic of a lymph node (biorender.com) with legends indicating capsule, node cortex and medulla and follicles. **Middle,** histological section of a mouse inguinal lymph node (4 μm thick) stained with hematoxylin-eosin, Bar = 250 μm. **Right,** inguinal lymph node of a non-infected Flk1-GFP mouse. Image is reconstructed from 4 panels (2x2), each panel corresponding to the maximum of intensity (MIP) calculated on a 150 μm section of the bubo. Bar = 250 μm. **B.** Draining inguinal lymph node of a *Y. pestis*-infected Flk1-GFP mouse on D3 post infection. Lymph node is representative of features observed on five independent challenged mice. Image is reconstructed from 8 panels (2x4), each panel corresponding to the maximum of intensity (MIP) calculated on a 150 μm section of the bubo (15 sections separated by 10um). The red color corresponds to bacteria visible in the subcapsular sinus and deeper in the lymph node parenchyma. The blood vasculature is colored in green. Frame 1 correspond to magnification on S1 Video; frame 2 correspond to magnification on S2 Video; frame 3 correspond to panel C and S3 Video. The capsule is indicated with white arrows, the medulla with a white star, and the cortex with white crosses. Bar = 500 μm. **C.** The left panel shows the magnification of the central region of panel B (MIP calculated on 20μm, 40 sections separated by 0.5um). The right panel displays an interpretation of the picture: the blood vessel is discontinued in close proximity of groups of bacteria and individual cells or remnants of cells are surrounded by bacteria. Made with biorender. Bar = 10 μm.

[11,12,18,19,22]. On D3, in the RFP-CO92-infected Flk1-GFP mice model, masses of bacteria (in red) were visible in the subcapsular sinus compartment, with bacteria infiltrating the node parenchyma (Fig 1B). These observations corresponded to the classical pattern of infection of the draining lymph node through the lymph vascular system (S1 Fig) [11,12]. In the regions of the infected-node distant from the bacteria, the green staining of the vascular system was as

intense as in non-infected mice (Fig 1A, right panel). However, this fluorescence appeared weaker or missing in the areas where bacteria were visible (Fig 1B, S1 and S2 Videos). There was little or no superposition of green and red fluorescence on the same field. This phenomenon was observed in all sections examined (corresponding to the complete draining lymph nodes of five infected Flk1-GFP mice). A higher magnification of a 20 μm thick section of the parenchymal zone of the lymph node, allowing a 3D reconstruction of the zone, showed blood vessels that appeared degraded in the vicinity of bacteria (Fig 1C and S3 Video). In this field, what was interpreted as an interrupted vessel was observed next to round single cells or degraded cell remnants. The confocal microscopy allowed to investigate the fields above and below the ones presented. As opposed to the left part of the vessels (Fig 1C) that disappear out of the field (partially visible on the S3 Video), the vessel at the center is abruptly discontinued. There was no trace of the vessel in the space above or below the field presented, but instead round shapes of 10μm of diameter stained in green, strongly evocative of round endothelial cells. The absence of superposition of green and red fluorescence below the subcapsular sinus where bacteria were visible suggests that blood vessels were either repelled by the mass of bacteria, or destroyed by the invading bacteria as showed in the higher magnification. In infected draining lymph nodes at a more advanced state of infection, the red masse of bacteria eventually takes over and fill the whole bubo. Remnant of green blood vessels are scarce (S2A and S2B Fig). These results suggest that *Y. pestis* can disrupt blood vessels from the parenchyma of lymph nodes.

### *Y. pestis* disseminate via the bloodstream and cross the vascular barrier

We assume that the direct degradation of blood vessels by the bacteria provokes local hemorrhages that could allow bacteria to enter the bloodstream. Once in the bloodstream, *Y. pestis* disseminates and colonize other lymphoid organs (spleen, liver, secondary lymph nodes) [11,26], suggesting that bacteria have the capacity to translocate from the blood vessel lumen to the parenchyma of these organs, to form secondary infectious foci.

The presence of bacteria in the blood vessels of the spleen on D3 post-infection was confirmed by confocal microscopy (Fig 2) and cfu counting (an average of $6.10^4$cfu/organ). Contrary to the draining lymph node, colonized via the afferent lymph, where it was not possible to observe superposition of bacteria and blood vessels staining, the spleen of infected Flk-1 mice presented clear superposition of red and green (appearing yellow) demonstrating the presence of bacteria within the vessels. Bacteria were visible superposed to the blood vessels surrounding the white pulp of the spleen.

Similarly, in secondary infected lymph nodes (i.e. left inguinal lymph nodes that were not drained by the lymphatic flux coming from the right primary draining lymph node) dense spots of bacteria were frequently observed inside blood vessel lumens (Figs 3 and S3). These spots were observable in the cortex of the lymph node, distant from the subcapsular sinus, consistent with a penetration of the bacteria in the lymph node via the bloodstream rather than via lymphatic flux. Higher magnification confirmed the presence of red bacteria inside the blood vessel lumen (Fig 3B and S4 Video). Furthermore, at this magnification we observed bacteria in the process of leaking out of the vessel into the lymph node parenchyma, strengthening the premise that *Y. pestis* has the capacity to degrade the blood barrier from the lumen to penetrate into the organ. In infected lymph nodes at a more advanced state of infection, the red spot of bacteria spread from the initial blood vessel through the cortex (S2C Fig). Similarly to the draining lymph nodes, blood vessels within the cortex seem to disappear while masse of bacteria progresses. Altogether, these data show that *Y. pestis* colonizes secondary organs by penetrating through the blood circulation and subsequently by degrading the blood vessel barrier.

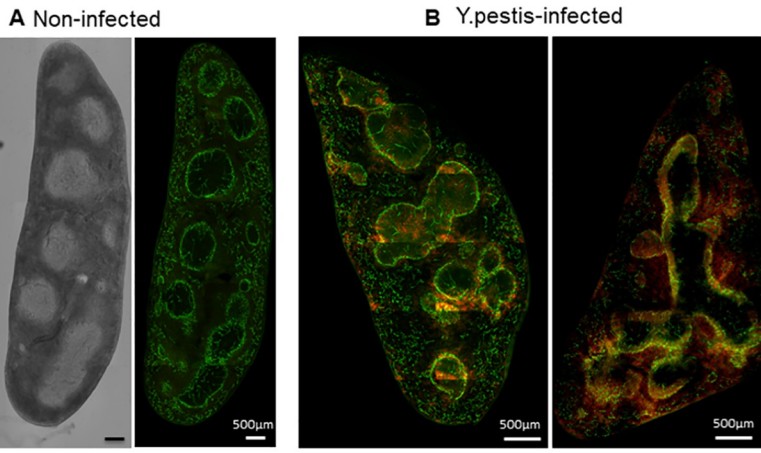

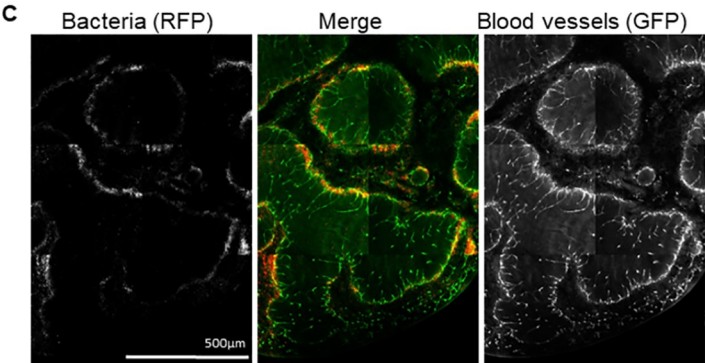

**Fig 2. Confocal microscopy observation of blood vessels in the spleen of non-infected and *Y. pestis*-infected Flk1-GFP mice.** For all images, the red color corresponds to bacteria and the blood vasculature is colored in green. **A.** Non-infected Flk1-GFP mouse spleen. Left, white field picture showing the visual structure of the spleen with white pulp appearing clear grey and red pulp appearing darker. Right, organization of blood vasculature in the red pulp and around the white pulp. **B.** Spleen from two *Y. pestis*-infected Flk1-GFP mice on D3 post-infection. Clear presence of red *Y. pestis* bacteria in blood vasculature, particularly around the white pulp. On the left panel blood vessels around white pulp appear strongly disorganized and partially disrupted. **C.** *Y. pestis*-infected mouse on D3 post-infection. Bacteria (left panel) and blood vessels (right panel) are merged in the middle panel where bacteria fluoresce in red and blood vessels in green. Images are reconstructed from panels each corresponding to the maximum of intensity (MIP) calculated on 200 μm thick sections (A: 10 panels (2x5); B left: 24 panels (4x6); B right: 18 panels (3x6); C: 6 panels (2x3) calculated on 150 μm thick sections). Bar = 500 μm.

## *Y. pestis* disrupts confluent monolayers of microvascular endothelial cells

An *in vitro* model was established to further investigate the interaction between *Y. pestis* and blood vessels using confluent monolayers of Human Dermal Microvascular Endothelial Cells (HDMEC). While non-infected HDMEC formed a confluent monolayer with tight junctions (Fig 4A), holes were visible at the junction between cells after 2.5h of infection with *Y. pestis* CO92 at an MOI of 100 (Fig 4B). These holes were observed around approximately 40% of the cells, and only around cells that were directly in contact with bacteria (Fig 4C), suggesting that a direct interaction between the cells and the bacteria was needed to induce this effect. This hypothesis was further examined by incubating HDMEC monolayers for 5h with either i) the filtrate of the supernatant of *Y. pestis* cultured at 28˚C or 37˚C for 3h or ii) with a sonicate of $10^7$ bacteria cultured at 28˚C or 37˚C for 3h. No holes were observed between cells in either case (S4 Fig). This result strengthens the premise that no toxins or bacterial cytoplasmic

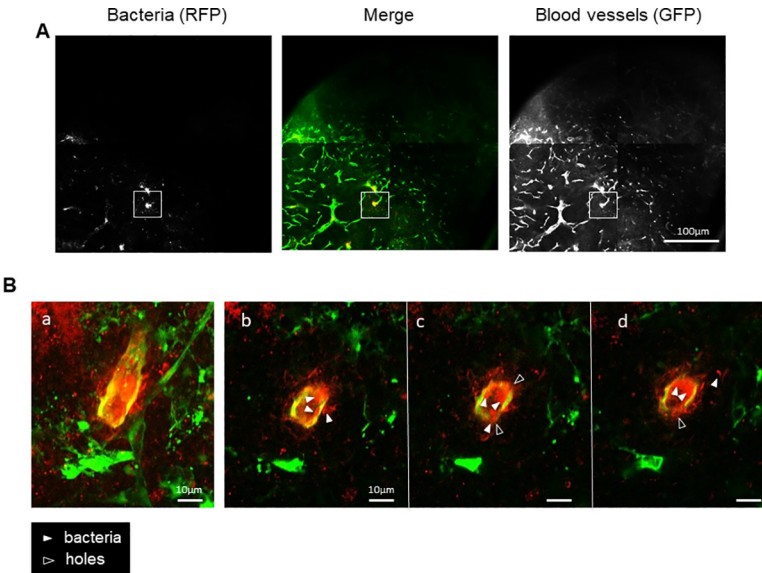

**Fig 3. Confocal microscopy, observation of the blood vessels and bacteria in the non-draining lymph node of a *Y. pestis*-infected Flk1-GFP mouse. A.** Bacteria (left panel) and blood vessels (right panel) are merged in the middle panel. Bar = 100 µm. Image are reconstructed from 4 panels (2x2), each panel corresponding to the maximum of intensity (MIP) calculated on a 200 µm section of the bubo (20 sections separated by 10um). Frame correspond to panel B and S4 Video. **B.** Magnifications of a blood vessel filled with bacteria. The panel **a.** represent the MIP calculated on a 50 µm thick section (100 sections separated by 0.5um); panels **b-c-d** are successive sections of panel **a.** White plain arrowheads point at red bacteria within and outside the vessel, white empty arrowheads point at some holes in the vessel. Bar = 10 µm.

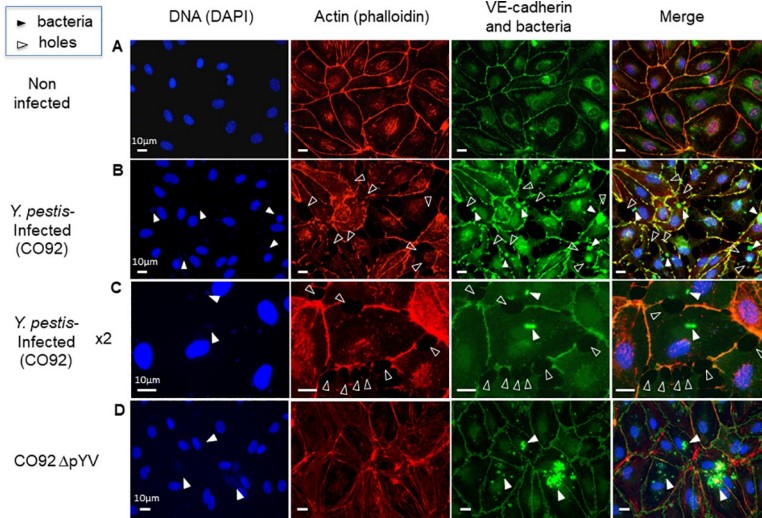

**Fig 4. Disruption of a monolayer of vascular endothelial cells by *Y. pestis*.** HDMEC monolayers were either uninfected (**A**, control), infected for 2.5h with the CO92 strain of *Y. pestis* at a MOI of 100 (**B** and **C**) or with a CO92 strain cured of the pYV/pCD1 plasmid (**D**); Panel **C** is a 2x magnification for better visualization. bars = 10 µm. The cells were fixed and stained to visualize their DNA (DAPI, blue), actin (phalloidin, red), and tight junctions (anti-VE-cadherin antibody, green). Bacterial cells on the monolayers were stained with an antibody directed against the F1 pseudocapsule (green) and their DNA was labeled with DAPI. Holes at the cell junctions were observed in the monolayers infected with *Y. pestis* (white empty arrowheads point at some holes in panels B and C). Contrast on the green panel B was enhanced to improve visualization of the holes. Bacterial DNA (DAPI) and *Y. pestis* were visible on the HDMEC monolayer at high magnification (white plain arrowheads).

molecules secreted outside the host cell cytoplasm were directly responsible for the observed phenotype and that a direct contact of live bacteria with the cells was necessary. The number of holes increased over time: after 8h of infection most cells were infected and started to separate from adjacent cells, and at 24h post-infection all cells were completely round (S5D Fig) and started to detach from the surface of the well.

### The pYV/pCD1 plasmid is necessary to increase the permeability and cross the vascular barrier

Since a direct interaction between bacteria and cells seemed necessary for *Y. pestis* to disrupt the vascular endothelial cell monolayer, the role of three major virulence factors exposed at the surface of the bacillus was investigated. First, the role of the plasminogen activator Pla carried by the pPla plasmid and the F1 pseudocapsule carried on the pMT plasmid of *Y. pestis* were tested [25]. Both, the derivative strain of *Y. pestis* CO92 in which the *caf* operon was deleted by allelic exchange (Δ*caf*) [28] and the 6/69 *Y. pestis* strain cured of the pPla plasmid (ΔpPla) [27] retained the same ability as the parental strains to cause holes at the cell junctions (S5 Fig). Second, a pYV/pCD1-cured derivative of strain CO92 was used to infect confluent HDMEC. As shown in Fig 4D, when infected with the pYV/pCD1-cured *Y. pestis*, no holes were observed in the HDMEC monolayer despite the presence of numerous bacteria in contact with the cells.

To quantify this phenomenon, we measured the permeability of the vascular barrier following infection using a Transwell system (Fig 5). The non-infected monolayer showed little permeability to FITC-Dextran ($<0,1$ug/min/cm$^2$), while the confluent monolayer became permeable over time after addition of Mannitol ($2,3$ ug/min/cm$^2$ after 3h), a hyperosmolar osmotic molecule known to induce the opening of cellular tight junctions. The infection with *Y. pestis* induces an even stronger increase in monolayer permeability after 3h of infection ($4,18$ ug/min/cm$^2$). The permeability of the monolayer did not increase when infected with the pYV/pCD1- derivative strain (Fig 5A and 5B). In addition, the increased permeability of the endothelial cells barrier allowed the passage of bacteria through the monolayer in a pYV/pCD1 dependent manner (Fig 5C). Altogether, these data demonstrate the essential role of the pYV/pCD1 plasmid in the increased permeability of blood vessels measured *in vitro* and observed *in vivo* during *Y. pestis* infection. According to these results, the ability of *Y. pestis* to enter and leave the blood circulation, thus invading the internal organs of its host and eventually provoking bleedings, can be explained by the deterioration of the vascular barrier carried by the functions encoded on the pYV/pCD1 plasmid.

## Discussion

In order to study the interaction between *Y. pestis* and its host vasculature directly *in vivo*, a new approach was developed using a mouse strain expressing GFP-tagged blood vessels infected with a RFP-tagged *Y. pestis*. This technology permitted an innovative, qualitative observation of infection and of the interaction between *Y. pestis* and its host at the organ level (using spinning-disc confocal microscopy) and at the cellular level (using confocal microscopy). The use of a spinning-disc confocal microscopy allowed analyzing the entire lymphoid organ (3x1.5 millimeters long and up to a millimeter of thickness) with a minimal transformation and alteration of the sample due to staining or treatment. In this study, this approach showed the previously undocumented degradation of the blood vessels by *Y. pestis* during infection. We propose that the degradation of blood vessels is an active mechanism caused directly by *Y. pestis* as observed *in vitro*.

An alternative explanation would be an indirect effect of *Y. pestis* on blood vasculature through the inflammatory response to the infection. Massive recruitment of Polynuclear

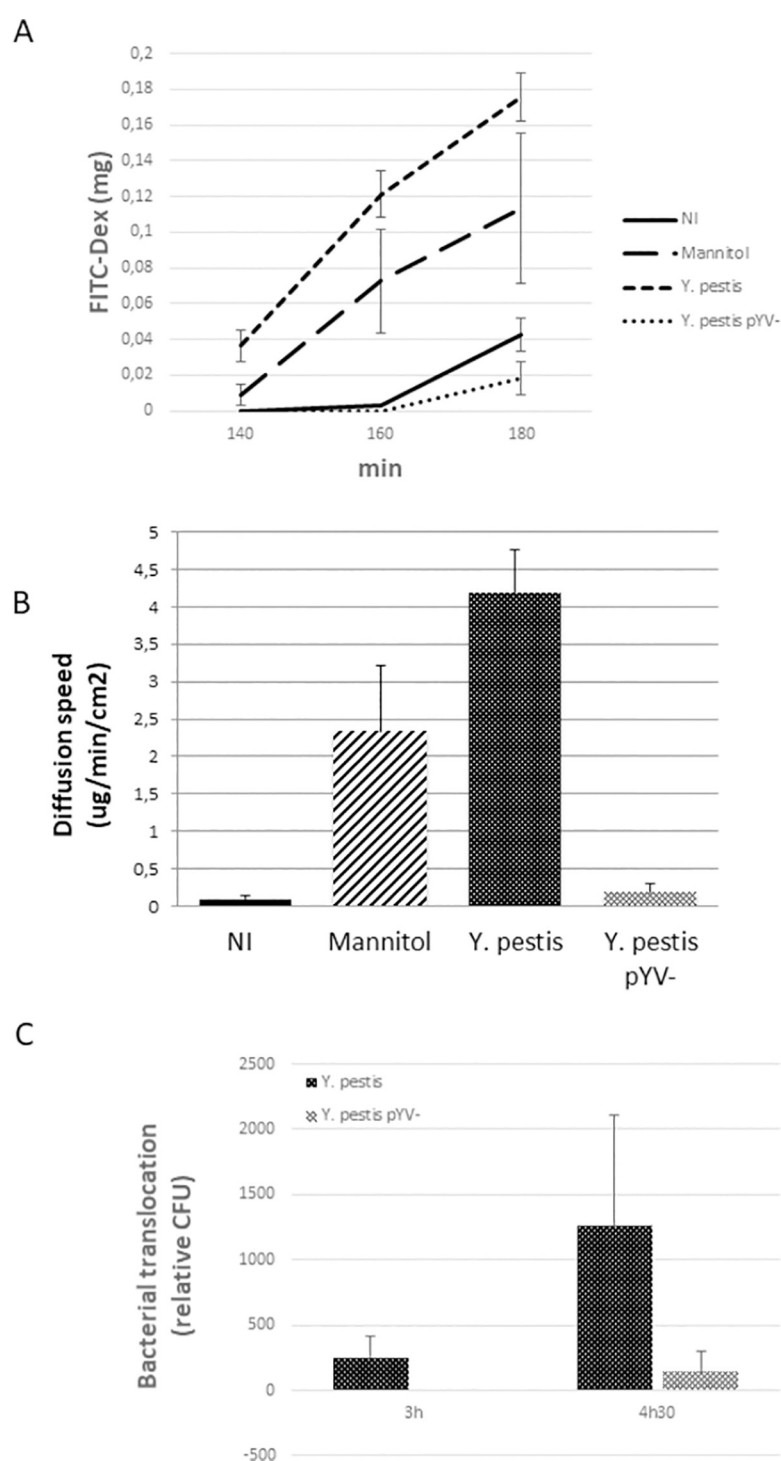

**Fig 5. Permeability of *Y. pestis*-infected HDMEC.** HDMEC monolayers cultured on Transwells (filter 0.4 μm) were infected (MOI = 100) for 2.5h with *Y. pestis* CO92 or its pYV/pCD1 cured derivative. Permeability was measured using FITC-Dex (4kDa) (**A**) and the diffusion speed calculated (**B**). Mannitol was used as positive control for junction opening. **C.** The crossing of bacteria was measured using Transwells of lower filter capacity (3 μm).

Neutrophils is known to provoke an increase in vascular permeability that could lead to bleeding in some cases [36]. However, neutrophils recruitment does not induce a complete physical degradation of the vessels themselves, such as the one observed in the infected lymph nodes. Additionally, an inflammatory process would have a distant effect, nevertheless, Fig 1 shows that only blood vessels in close proximity with bacteria seem to disappear, suggesting that it is the direct contact of *Y. pestis* with endothelial cells that provokes the degradation of the vessels, as seen *in vitro*, and thus excluding a major role of the inflammatory process in this phenomenon.

This point is of importance since the mechanism used to enter the bloodstream remains controversial. Previous work by Nahm and colleagues showed that the variable timing between the subcutaneous infection of *Y. pestis* and the outcome is due to the time to reach the lymph node [10]. Once in the draining lymph node, *Y. pestis* quickly spread to other internal organs, the infection is rapid and often fatal. Thus suggesting that lymph nodes are the entry door to blood circulation. The main argument against the hypothesis for a direct entry of the bacillus into the bloodstream was the absence of proof that *Y. pestis* enters in contact with vessels and could cross the vascular barrier. This work demonstrates that both events take place *in vivo* in the bubo. Though it does not invalidate other hypotheses, the data presented strongly support for an active and direct entry of *Y. pestis*, in a pYV/pCD1 dependent manner, into the blood circulation and explains the invasiveness of this pathogen. The increased blood vessel permeability allows the crossing of bacteria, which could also explain systemic hemorrhages associated with plague. Past studies suggested that hemorrhages due to *Y. pestis* were the consequences of a DIC provoked by the toxicity of the bacterial LPS [18,20]. However, this work suggests a more direct and faster effect of the bacteria on blood vasculature. *Y. pestis* can breach the blood barrier *in vitro* and *in vivo* inducing an increased permeability that could provoke internal bleedings.

Among the virulence factors carried by *Y. pestis*, the pYV/pCD1 plasmid is one genetic element crucial to the pathogenicity [25]. This ≈70 kb plasmid encodes a TTSS that mediates the injection of molecular effectors, the Yop proteins, directly into the cell cytoplasm. Inside the cells, these Yops disorganize the actin cytoskeleton (thus preventing phagocytosis), inhibit signaling cascades involved in the early innate response, and induce apoptosis of the target cell [26,37,38]. We propose that the pYV/pCD1 plasmid is responsible for the disruption of blood vessels, probably through a direct action of the Yop effectors injected into the cytoplasm of endothelial cells. The fact that neither the bacterial sonicate nor the bacterial supernatant can produce the rounding phenotype observed during infection suggests that the Yop effectors themselves cannot act from outside the eukaryotic cells. We assume that the injection into the cytoplasm is necessary for inducing the rounding. The YopT and YopE effectors are good candidates as they have been demonstrated to have a rounding capacity on Hela cells [39,40,41]. The TTSS has been showed to be crucial in *Y. pestis* pathogenicity and a deletion mutant of pYV/pCD1 is virtually non-pathogenic. In particular, a pYV/pCD1- strain injected by IV route is almost non-pathogenic. We propose that the pYV/pCD1 plasmid encoding the TTSS is responsible for the impressive invasiveness of *Y. pestis* and for its hemorrhagic features.

## Supporting information

**S1 Video. Magnification 1 of confocal microscopy observation of blood vessels in the bubo of *Y. pestis*-infected Flk1-GFP mice.** Draining inguinal lymph node of a *Y. pestis*-infected Flk1-GFP mouse on D3 post infection. Magnification of the frame 1 from Fig 1A. Video shows 15 successive panels separated by 10um (from top to bottom) used to reconstruct the MIP of Fig 1A. The red color corresponds to bacteria visible in the subcapsular sinus and deeper in

the lymph node parenchyma. The blood vasculature is colored in green. There is little to no superposition of the staining.
(AVI)

**S2 Video. Magnification 2 of confocal microscopy observation of blood vessels in the bubo of *Y. pestis*-infected Flk1-GFP mice.** Draining inguinal lymph node of a *Y. pestis*-infected Flk1-GFP mouse on D3 post infection. Magnification of the frame 2 from Fig 1A. Video shows 15 successive panels separated by 10um (from top to bottom) used to reconstruct the MIP of Fig 1A. The red color corresponds to bacteria visible in the subcapsular sinus and deeper in the lymph node parenchyma. The blood vasculature is colored in green. There is little to no superposition of the staining.
(AVI)

**S3 Video. Magnification 3 of confocal microscopy observation of blood vessels in the bubo of *Y. pestis*-infected Flk1-GFP mice.** Draining inguinal lymph node of a *Y. pestis*-infected Flk1-GFP mouse on D3 post infection. Magnification of the frame 3 from Fig 1A. Video shows 26 successive panels separated by 0.5um (from top to bottom) used to reconstruct the MIP of Fig 1C. The red color corresponds to bacteria and blood vasculature is colored in green. Individual bacteria are identifiable. The continuous blood vessel in green on the right of the image appears degraded and single cells are visible on the left next to bacteria.
(AVI)

**S4 Video. Magnification 4 of confocal microscopy observation of blood vessels in the secondary lymph node of *Y. pestis*-infected Flk1-GFP mice.** Non-draining inguinal lymph node of a *Y. pestis*-infected Flk1-GFP mouse on D3 post infection. Magnification of the frame from Fig 2A. Video is a 3D reconstruct corresponding to the MIP showed on Fig 2B.a. The red color corresponds to bacteria and blood vasculature is colored in green. The continuous blood vessel in the middle of the reconstruction appears full of red bacteria. Superposition of staining green and red (appearing yellow) is observable. Individual bacteria are identifiable inside and leaking outside the vessel.
(AVI)

**S1 Fig. Histopathological observation of an *Y. pestis*-infected draining lymph node of ORF1 mice.** Histological section of a mouse inguinal lymph node (4 μm thick) stained with hematoxylin-eosin. ORF1 mouse was infected subcutaneously with 500cfu CO92 *Y. pestis* strain for 76h prior to sacrifice. **A.** Full section of the lymph node. Large infiltrate of *Y. pestis*, colored in pink, are visible at the periphery of the lymph node (black arrowheads), progressing from the lymphatic sinus to the cortex. Edemas and large engorged blood vessels (erythrocytes are colored in red) are visible within the lymph node. Infiltrate of Polynuclear neutrophils (PMN) with the characteristic horseshoe shaped nuclei colored in blue/purple are visible in the cortex. Bar = 500 μm **B.** Higher magnification of the square B displaying a "sea" of bacteria (black arrowheads, black frame surrounding pink areas) and PMNs infiltrates around the bacteria (yellow five-branch stars). Hemorrhages are visible in the tissue surrounding the lymph node (six-branch red stars). **C.** Higher magnification of the square C displaying a "sea" of bacteria (black arrowheads), PMNs infiltrates (yellow five-branch stars) and enlarged engorged blood vessels (red). A hemorrhage is visible in the cortex of the lymph node (six-branch red stars). Bacteria are visible are close proximity of the blood vessel. Blood vessels are tampered, but not fully degraded.
(TIF)

**S2 Fig. Confocal microscopy observation of blood vessels in advanced draining lymph node and secondary lymph node of *Y. pestis*-infected Flk1-GFP mice.** Inguinal draining lymph nodes and secondary lymph node of *Y. pestis*-infected mice displaying an advanced state of degradation on D3 post-infection. The images are reconstructed from panels corresponding to the maximum of intensity (MIP) calculated on 150–200 um sections. Red and green panels are merged. **A**. and **B**. are two consecutive slices of the same draining lymph node. Red bacteria are filling the entire volume of the cortex, all vasculature within the cortex seems destroyed. **C**. is a secondary lymph node infected through blood circulation. Dense spots of bacteria spread through the cortex from the initial entry point. Blood vessels within the cortex have progressively disappeared on the side of numerous bacterial areas (top half) compared to the other side (bottom half).
(TIF)

**S3 Fig. Confocal microscopy observation of blood vessels in the secondary lymph node of additional *Y. pestis*-infected Flk1-GFP mice.** Secondary lymph nodes of 5 *Y. pestis*-infected mice on D3 post-infection. The images are reconstructed from panels corresponding to the maximum of intensity (MIP) calculated on 150–200 μm sections. Bacteria are tagged with RFP and appeared as red spots co-localizing with GFP-tagged blood vessels (arrowheads). Red and green panels are merged. The average size of a lymph node is 1 to 2 mm.
(TIF)

**S4 Fig. Monolayer of HDMEC cells treated with *Y. pestis* sonicate and filtrate.** HDMEC monolayers were incubated for 5h with the equivalent of $10^7$ bacteria *Y. pestis* culture medium (filtrate) or $10^7$ bacteria sonicate (see Materials and Methods section). No holes in the tight junctions are observed due to these treatments. DNA was stained with DAPI (blue; panels a,e, i). Actin was stained with Phalloidin (Red; panels b,f,j). Tight junctions were targeted with an anti-VE-cadherin antibody (green, panels c,g,k).(d,h,l) are merged panels. Bar = 10 μm.
(TIF)

**S5 Fig. Infection of a monolayer of HDMEC cells with various *Y. pestis* derivatives.** HDMEC monolayers were infected (MOI = 100) for 2.5h with *Y. pestis* 6/69 wild type (**A**), its derivative cured of pPla (**B**), and *Y. pestis* CO92 Δ*caf* (**C**). **D.** HDMEC monolayers were infected (MOI = 100) for 24h with *Y. pestis* CO92. DNA was stained with DAPI (blue; panels a,e,i,m). Actin was stained with Phalloidin (Red; panels b,f,j,n). Staining was done as described in the legend of Fig 4. Examples of bacteria in close contact with the cells are indicated with white plain arrowheads. Bar = 10 μm.
(TIF)

## Acknowledgments

The authors would like to thank Petra Dersch from the department of Molecular Infection Biology, Helmholtz Centre for Infection Research, Braunschweig, in Germany, for providing the plasmid *pFU96* and Anne Derbise, in the Yersinia research unit for her benevolent assistance in constructing the *RFP-CO92* strain. We are very grateful to Fabrice Chrétien and Claire Latroche from the Histopathology and Animal Models Unit at the Institut Pasteur, Paris, France, for providing the *Flk1-GFP* mice. We would like to thank Christian Demeure for his critical reading of the manuscript and with Sofia Filali from in the Yersinia research Unit for their helpful support with animal experiments over the years.

## Author Contributions

**Conceptualization:** Guillain Mikaty, Elisabeth Carniel.

**Data curation:** Guillain Mikaty.

**Formal analysis:** Guillain Mikaty, Laurence Fiette.

**Funding acquisition:** Elisabeth Carniel.

**Investigation:** Guillain Mikaty, Héloïse Coullon, Laurence Fiette.

**Methodology:** Guillain Mikaty.

**Project administration:** Guillain Mikaty, Elisabeth Carniel.

**Supervision:** Elisabeth Carniel.

**Validation:** Elisabeth Carniel.

**Writing – original draft:** Guillain Mikaty.

**Writing – review & editing:** Guillain Mikaty, Javier Pizarro-Cerdá, Elisabeth Carniel.

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
