## [Decision Letter · Decision Letter 0]

23 Jul 2021

Dear Dr. Mikaty,

Thank you very much for submitting your manuscript "The invasive pathogen Yersinia pestis disrupts host blood vasculature to spread and provoke hemorrhages." for consideration at PLOS Neglected Tropical Diseases. As with all papers reviewed by the journal, your manuscript was reviewed by members of the editorial board and by several independent reviewers. In light of the reviews (below this email), we would like to invite the resubmission of a significantly-revised version that takes into account the reviewers' comments. 

I have received the reviews for your manuscript. The reviewers have evaluated your manuscript and find it valuable to the field. However, they raise a few important questions. Additional data is needed to resolve them. A fully revised manuscript will be evaluated before publication in PLOS NTD.

We cannot make any decision about publication until we have seen the revised manuscript and your response to the reviewers' comments. Your revised manuscript is also likely to be sent to reviewers for further evaluation.

Sincerely,

R. Manjunatha Kini

Associate Editor

Fabiano Oliveira

Deputy Editor

I have received the reviews for your manuscript. The reviewers have evaluated your manuscript and find it valuable to the field. However, they raise a few important questions. Additional data is needed to resolve them. A fully revised manuscript will be evaluated before publication in PLOS NTD.

Reviewer's Responses to Questions

**Key Review Criteria Required for Acceptance?**

**Methods**

-Are the objectives of the study clearly articulated with a clear testable hypothesis stated?

-Is the study design appropriate to address the stated objectives?

-Is the population clearly described and appropriate for the hypothesis being tested?

-Is the sample size sufficient to ensure adequate power to address the hypothesis being tested?

-Were correct statistical analysis used to support conclusions?

-Are there concerns about ethical or regulatory requirements being met?

Reviewer #1: See my comments below

Reviewer #2: Methods were appropriate to ask the relevant questions.

Reviewer #3: Methods are appropriately described.

**Results**

-Does the analysis presented match the analysis plan?

-Are the results clearly and completely presented?

-Are the figures (Tables, Images) of sufficient quality for clarity?

Reviewer #1: See my comments below

Reviewer #2: The results are presented clearly and concisely, and Figures are of high quality.

Reviewer #3: Results are clearly described.

**Conclusions**

-Are the conclusions supported by the data presented?

-Are the limitations of analysis clearly described?

-Do the authors discuss how these data can be helpful to advance our understanding of the topic under study?

-Is public health relevance addressed?

Reviewer #1: Major Concerns:

1) The authors’ main conclusion in this paper is that Y. pestis, via a T3SS-dependent mechanism, actively damages the blood vessel endothelium, thus increasing vessel permeability, and this is how the bacteria enter the bloodstream and disseminate. I do not feel that the data the authors present here allow them to rule out the other 2 potential mechanisms of dissemination to the blood described by the authors in the introduction (movement of bacteria through the lymph and eventually into bloodstream, or carriage of bacteria into the blood by phagocytes). I can see how damage to the vessels would cause leakage out of the vessels, but that does not necessarily mean bacteria would enter those damaged vessels. 

2) At the time point chosen by the authors for their in vivo experiments, there are really two different phenomena being studied regarding Y. pestis dissemination and it is a bit difficult to separate the two. The first relates to how the bacteria are entering the bloodstream. The second relates to how these disseminated bacteria move from the blood into the tissue parenchyma. For the first, since the bacteria would initially be found in the subcapsular sinus area early in infection, the bacteria would need to gain access to the LN parenchyma to come in direct contact with blood vessels to increase their permeability. The authors do not address how the bacteria might be getting to the parenchyma initially. Because of the later time point (d3), the blood vessel damage the authors see in the dLN could have been caused from the inside out by the bacteria already spreading systemically, rather than the bacteria that arrived in the LN via afferent lymph from the injection site. For the second, it’s easier to see how the bacteria already in the bloodstream could damage the vessels and leak out into the parenchyma and cause the hemorrhages the authors discuss. I feel that this study would benefit from the addition of earlier time points for the in vivo work. Being able to show the early stages of blood vessel damage, or at least bacteria in contact with vessels and able to inject Type III effectors, prior to dissemination of the bacteria throughout the body, would greatly strengthen the support for the authors’ conclusions.

3) The authors do not address the potential role of inflammatory cells, particularly neutrophils, could be playing in the blood vessel damage they observe. At the later time point they focus on, there would potentially be large numbers of neutrophils recruited to the draining LN and these cells would likely colocalize with areas of high bacterial density, and therefore, areas of blood vessel damage. The possible role of the inflammatory response in the observed phenomena should at least be discussed, if not assessed experimentally.

4) For the in vitro experiments, the authors conclude that the Yops must be secreted into the HDMEC cells by the T3SS. This is probably a reasonable assumption, but methods are available for confirming the secretion of Yops into host cell cytosol. Determining the specific Yop or Yops responsible would also be fairly straightforward.

Reviewer #2: The authors do a good job of contextualizing the findings and highlighting the importance to understanding bubonic plague. The conclusions are largely supported by the data. There are some inherent limitations to the microscopy, even though it is the most striking and compelling aspect of the manuscript. For instance, it is suggested that disruption of blood vessels is visualized- and that is likely the case, though it is only visual analysis and there is no conclusive evidence that it is what we are seeing (ie. control with agent that causes disruption).

Reviewer #3: The conclusions are not fully supported by the data. Some experiments are missing that are required for the conclusions made, or alternative hypotheses needed to be included.

**Editorial and Data Presentation Modifications?**

Reviewer #1: Line 264-277: This section refers to a supplementary figure. It seems like the figure could easily be included as regular figure in the text.

Line 274: The authors refer to “secondary organs”, but I think they can specify spleen here.

Figure 2B: I think this figure would benefit from the addition of arrows pointing to the phenomena being described.

Line 308: The authors refer to “holes in cells”, but the holes seem to be gaps in the monolayer between cells at the junctions. Thus, it is not clear how the “holes were observed in approximately 40% of the cells” was calculated

Figure 4A: It’s unclear why the Y-axis is in milligrams. The methods implied that the fluorescence of FITC-dextran was what was being measured. Is the fluorescence measurement being converted to mg? If so, shouldn’t it be expressed as a concentration instead of just total mg? 

Some suggestions for improving the text:

Line 33: change to “…remain poorly understood.”

Line 38: change to “…plasmid is responsible for…”

Line 45: “It is among the most important bacterial…”

Line 59: change to “Y. pestis can multiply within lymph nodes. If the innate immune response in the lymph node is strong…”

Line 74: “…have been a striking feature…”

Line 84: “…factors that play…”

Line 89: “…activities capable of converting plasminogen to plasmin, thus degrading extracellular matrix and fibrin clots in vivo.” 

Line 94: “…associated with and essential for the virulence…”

Line 98: “…may be important during…”

Line 200: delete the sentence starting “This RFP CO92…”. Redundant

Line 211: “…shows the highly vascularized medulla…”

Line 271: “…and the classical cfu…”

Line 272: “Contrary to the draining lymph node, colonized via the afferent lymph,”

Line 285: “…at this magnification we observed bacteria in the process of leaking out of the vessel into the lymph node parenchyma…”

Line 314: “…cells in either case…”

Line 315: “no toxins or bacterial…”

Line 316: “…host cell cytoplasm…”

Line 387: “…increased blood vessel permeability allows the crossing of bacteria, which could also explain systemic hemorrhages associated with plague.”

Line 392: “Y. pestis can breach…permeability that could provoke…”

Line 402: “…supernatant can produce…”

Reviewer #2: Minor concerns/comments:

Of note, pCD1 is the more common nomenclature for the Y. pestis plasmid as opposed to pYV which is typically used to denote the plasmind from Y. pseudotuberculosis or Y. enterocolitica.

The sentence beginning in 38 should likely read "pYV plasmid bears responsibility" or pYV plasmid is responsible"

Line 73- may want to slightly expand on what "bleedings" refers to specificlly (ie. internal hemorrhage?)

Lines 77-79- it is a little confusing as to how DIC (which is increased coagulation) could be responsible for hemhorage, may want to slightly clarify

Line 94- specify "inside the host cell cytosol"

Lines 94-96 state that the three plasmids are essential to virulence of the pathogenic Yersinia, but this is only true of Yersinia pestis- pCD1/pYV is the only plasmid of the 3 found in all three pathogenic Yersinia.

Introduction may benefit from 1 or 2 sentences summarizing data or conculsions. 

Line 187 italicize Y. pestis.

Lines 190-191 should read "mechanisms involved....were never properly determined" or "mechanism involved...was never properly determined)

The statement in line 201 "retained the same virulence" does not have any confirmative data except for the LD50, and should perhaps read "retained the same LD50" (unless bacterial burden and survival is to be included in the supplement)- obviously a very minor concern if at all.

Figure 1- Capsule appears to point to a follicle on the GFP panel, which may be confusing

It is difficult to determine if the images shown in Figure 1C are truly showing degraded cell remnants of vessels, as is suggested. Though this reviewer is not sure if it is possible, a control with an agent that disrupts the vasculature in some way or a marker of vascular damage would strengthen the argument.

On Page 12, it is indicated that bacteria are entering secondary infected lymph nodes through only the vasculature. Though likely, it is difficult to confirm that Figure 2B is showing "bacteria leaking to from the vessel into the parenchyma", it is not clear that those bacteria in the parenchyma originated in the vasculature vs entering the node in another manner (ie lymphatic system) and degrading vessels. This question isn't asked specifically (different timing, differential labeling, etc). Therefore language in this section with that regard should perhaps be tempered. 

The authors do not comment on whether or not they see any disruption of the vasulature in the non-draining lymph nodes in which the bacteria have assumedly arrived by blood vessels. Is it proposed that the disruption of blood vessels strictly occurs from "outside" of the vessel?

Supplementary Figure S2 is fairly striking, and might be included in the main text. 

In the discussion, it may be useful to comment on what it might mean if the vessel destruction is exclusive to bacteria that have arrived in the lymph nodes via the lymphatic system in the draining lymph node vs the destruction seen (or not seen) in the non-draining lymph node in which the bacteria have entered presumably via the vascular system.

Reviewer #3: See minor comments below.

**Summary and General Comments**

Reviewer #1: In their manuscript “The invasive pathogen Yersinia pestis disrupts host blood vasculature to spread and provoke hemorrhages” Mikaty et al. use in vivo and in vitro approaches to examine the effects of Y. pestis infection on the blood vessel endothelium. For the in vivo work, the authors used a mouse strain expressing GFP in blood vessel endothelial cells, combined with red-fluorescent Y. pestis, to image bacteria and vessels in fixed draining lymph nodes, non-draining lymph nodes, and spleen. For the in vitro work, the authors look at the effects of Y. pestis on the permeability of human endothelial cell monolayers.

Overall, the work is original and important, and the results are potentially very interesting. However, I have concerns about some of the authors’ conclusions that I feel need to be addressed. I also feel that the manuscript could use quite a bit of editing and I will suggest just some of the changes that could be made.

Reviewer #2: In this study, the authors utilize both in vivo and in vitro analysis to describe the pCD1-mediated degradation/destruction of the blood vessels in the primary draining lymph node after infection with Y. pestis. Using in vivo microscopy, they show presence of bacteria consistent with lymphatic vessel-mediate arrival in the draining lymph node, and clear disruption of blood vessels in the localized areas around the clusters of bacteria. The lack of co-localization between bacteria and vasculature in the draining lymph node is striking, as is the contrast between this finding and what is seen in the non-draining lymph node. It is proposed that this disruption/destruction of the vasculature is responsible for entry of the bacteria into the bloodstream and subsequent colonization of other lymphoid organs. The authors are able to show this in non-draining lymph nodes as well as the spleen. In the non-draining lymph nodes the authors see bacteria associated with the vasculature as well as within the parenchyma of the lymph nodes.

 Overall, the manuscript is well-written and the data clearly presented. The microscopy is beautiful, and the authors do a nice job of piecing together the events that occur upon entry of Y. pestis into the draining lymph node, and the events subsequent to this critical step. This work most certainly contributes to our understanding of how Y. pestis disseminates within the body. The strengths of the manuscript are in the innovative microscopic technique, as well as in the comparison of colocalization of bacterial and vasculature within draining lymph nodes vs. non-draining lymph nodes and other organs. A weakness is that the manuscript is primarily compelling microscopy, which limits some of the conclusions. For instance, though the it is indicated that vessel destruction is observed as green blots within the staining, it can't be known for sure that this is what we are seeing, and though it is assumed that bacteria have leaked from he blood vasculature into the parenchyma in the non-draining lymph nodes, this can't be confirmed for certain. It also might have been informative to have a sort of time course to observe draining and non-draining lymph nodes simultaneously at different time points (ie. earlier timepoints where we see small numbers of bacteria arriving at the draining lymph nodes vs. presumably no bacteria in the non-draining) to confirm the model proposed by the authors. Any weaknesses are offset by the fact that the microscopy in particular gives us clear insight into how Y. pestis behaves within the lymph nodes, and the authors do a great job of piecing together a narrative as to what is happening. AS a result, I consider the findings/observations to be highly significant to the field, and the manuscript paints a very compelling story, or at least lays the ground work for testing a model of dissemination.

Reviewer #3: Summary

Yersinia pestis is a highly invasive lymphotropic pathogen that must gain access to the blood in order to be effectively transmitted from the infected mammalian host to its insect vector. However, the mechanism(s) used by the bacterium to breach the lymph node to colonize the bloodstream are not clearly defined. Here Mikaty et al. describe a genetically modified mouse model, which expresses GFP primarily in endothelial cells, to visualize blood vasculature in the lymph nodes during bubonic plague. The authors use confocal microscopy to characterize the blood vessel architecture in the draining lymph node during Y. pestis infection, showing for the first time that blood vessels near Y. pestis appear to be damaged, possibly resulting in hemorrhaging. The authors also use primary human endothelial cells in cell culture to show that infection with Y. pestis results in what appears to be loss of tight junctions, and suggests that similar effects could occur in vivo, leading to vessel damage and access to the blood stream by the bacterium. Finally, the authors demonstrate that the absence of the pCD1 virulence plasmid inhibits the ability of Y. pestis to induce morphological changes in endothelial cells and permeability of cell monolayers in vitro. This study introduces an exciting new in vivo model to help us to better understand the mechanisms used by Y. pestis to breach the lymph node and colonize the blood.

Comments:

This authors provide us with a clear characterization of the Flk-1GFP/+ mouse model in the context of bubonic plague, and implicates the Ysc T3SS and Yop effectors in causing endothelial damage and dissemination. While I think this model provides investigators with a powerful tool to study vascular damage during plague, the manuscript lacks clear mechanistic studies to justify some of the conclusions that the authors make in the manuscript. These conclusions could be significantly strengthened by additional experiments and/or a more detailed discussion of the results and data from previous published studies.

1. It is not surprising that Y. pestis infection of endothelial cells in vitro results in cell rounding and death. There are ample reports that Y. pestis will inject effectors into almost any cell in monoculture, but cell-type specific targeting is observed in vivo that demonstrates that immune cells are preferentially targeted (independently demonstrated by Marketon, Mecsas, and Pechous labs). Therefore, these monocultures studies do not demonstrate that Y. pestis is specifically targeting endothelial cells in vivo. It is still possible that the immune response to the bacteria is leading to vascular leakage/damage in vivo that is required for immune cell recruitment, which is part of the formation of the bubo, and responsible for the localized vascular damage observed. This possibility is not properly discussed. In vivo targeting of endothelial cells in the lymph by Y. pestis using B-lactamase reporters would provide better support for the authors’ hypothesis. Without it, I am not sure intoxication vs. immune response hypotheses can be differentiated.

2. The MOI used in the cell culture studies were extremely high. Justification for these numbers should be included. Better yet, a dose response might be more effective approach.

3. Are the endothelial cells used in the in vitro studies polarized? Does this matter in the context of infection? Are the cells that round up targeted/injected by the T3SS? 

4. Is the response by the endothelial cells due to the action of the Yop effectors or a host response to the infection (e.g., inflammasome activation and proptosis)? Infection with a strain expression the T3SS needle and lacking the effectors would help to understand the molecular mechanisms responsible for the phenotypes observed.

5. Fig. 3: Not sure why both VE-cadherin and bacteria were both stained green. Is the increased colocalization of the green and red channels in 3B due to increased VE-cadherin at the cell membrane or are these bacteria? If bacteria, why different from 3D? The actin staining in CO92 pYV- (3D) also appears to be very different from all other samples. Is this a fair observation, or an artifact of the representative image shown?

Minor comments:

1. Line 29: Not sure what is meant by “a powerful pathogen”. Suggested rewording.

2. Yop effectors are not considered classical toxins. Suggest changing the use of “toxins” with “effectors”. 

3. Line 94: Only pCD1/pYV is conserved in all three Yersinia species, pPla and pMT are not in the enteric pathogens. Please revise tis sentence.

4. The reader would benefit from including the structural references to Fig. 1B to orient them to the LN structure similar to shown in Fig. 1A.

5. Line 273: I think the “lymphatic door” needs to clearly described – not sure this term is going to be widely understood. 

6. Line 277: I think your conclusions would be better supported if you include the co-localization channel. Yellow color does not show up well on these images.

7. Line 349: You should include a description of the results of your positive control (mannitol) first then the Y. pestis data.

PLOS authors have the option to publish the peer review history of their article (what does this mean?). If published, this will include your full peer review and any attached files.

Reviewer #1: No

Reviewer #2: No

Reviewer #3: No
---

## [Editor Report · Decision Letter 1]

22 Sep 2021

Dear Dr. Mikaty,

We are pleased to inform you that your manuscript 'The invasive pathogen Yersinia pestis disrupts host blood vasculature to spread and provoke hemorrhages.' has been provisionally accepted for publication in PLOS Neglected Tropical Diseases.

Best regards,

R. Manjunatha Kini

Associate Editor

Fabiano Oliveira

Deputy Editor

The authors have revised the manuscript keeping all the comments made by three reviewers. I also understand that the authors' rebuttal regarding the molecular mechanism studies as they are part of their ongoing study.

I am satisfied with their responses and the revisions.

---

## [Editor Report · Acceptance letter]

30 Sep 2021

Dear Dr. Mikaty,

We are delighted to inform you that your manuscript, "The invasive pathogen Yersinia pestis disrupts host blood vasculature to spread and provoke hemorrhages.," has been formally accepted for publication in PLOS Neglected Tropical Diseases.

Best regards,

Shaden Kamhawi

co-Editor-in-Chief

Paul Brindley

co-Editor-in-Chief
